# Accuracy, Efficiency, and Transferability of a Deep Learning Model for Mapping Retrogressive Thaw Slumps across the Canadian Arctic

**Lingcao Huang** [1,*] **, Trevor C. Lantz** [2] **, Robert H. Fraser** [3] **, Kristy F. Tiampo** [4] **, Michael J. Willis** [4] **and Kevin Schaefer** [5]

1 Earth Science and Observation Center, Cooperative Institute for Research in Environmental Sciences, University of Colorado Boulder, Boulder, CO 80309, USA

2 School of Environmental Studies, University of Victoria, Victoria, BC V8P 5C2, Canada; tlantz@uvic.ca

3 Canada Centre for Mapping and Earth Observation, Natural Resources Canada, 560 Rochester Street, Ottawa, ON K1S 5K2, Canada; robert.fraser@NRCan-RNCan.gc.ca

4 Cooperative Institute for Research in Environmental Sciences and Geological Sciences, University of Colorado Boulder, Boulder, CO 80309, USA; kristy.tiampo@colorado.edu (K.F.T.); mike.willis@colorado.edu (M.J.W.)

5 National Snow and Ice Data Center, Cooperative Institute for Research in Environmental Sciences, University of Colorado Boulder, Boulder, CO 80309, USA; kevin.schaefer@colorado.edu

\* Correspondence: lingcao.huang@colorado.edu

**Abstract:** Deep learning has been used for mapping retrogressive thaw slumps and other periglacial landforms but its application is still limited to local study areas. To understand the accuracy, efficiency, and transferability of a deep learning model (i.e., DeepLabv3+) when applied to large areas or multiple regions, we conducted several experiments using training data from three different regions across the Canadian Arctic. To overcome the main challenge of transferability, we used a generative adversarial network (GAN) called CycleGAN to produce new training data in an attempt to improve transferability. The results show that (1) data augmentation can improve the accuracy of the deep learning model but does not guarantee transferability, (2) it is necessary to choose a good combination of hyper-parameters (e.g., backbones and learning rate) to achieve an optimal trade-off between accuracy and efficiency, and (3) a GAN can significantly improve the transferability if the variation between source and target is dominated by color or general texture. Our results suggest that future mapping of retrogressive thaw slumps should prioritize the collection of training data from regions where a GAN cannot improve the transferability.

**Keywords:** DeepLab; domain adaptation; generative adversarial network; permafrost; thermokarst

## 1. Introduction

Increases in ground temperature and active layer thickness, coupled with elevated summer precipitation, are altering the frequency and size of disturbances initiated by permafrost thaw [1–4]. This intensification is concerning because thermokarst disturbances cause hydrological and geomorphic changes impacting northern communities, ecosystems, and global ecological processes [5–11]. Ice wedges in high Arctic uplands that have remained stable for millennia are being truncated by top down thaw, which causes ground subsidence and the formation of numerous thaw ponds [12,13]. In lake-rich lowland areas, thermokarst processes are increasing the frequency of rapid, irreversible lake drainage [14–16]. The rapid shift in climate in many Arctic regions has also caused a large increase in the area affected by retrogressive thaw slumps [3,17–19]. Thaw slumps develop in sloping terrain and consist of an ice-rich exposure or headwall and a downslope scar area (Figure 1). As ground ice in the headwall ablates, thawed materials are transported downslope and the scar area expands. Over annual to decadal time scales slumps can create horseshoe-shaped depressions in the landscape.

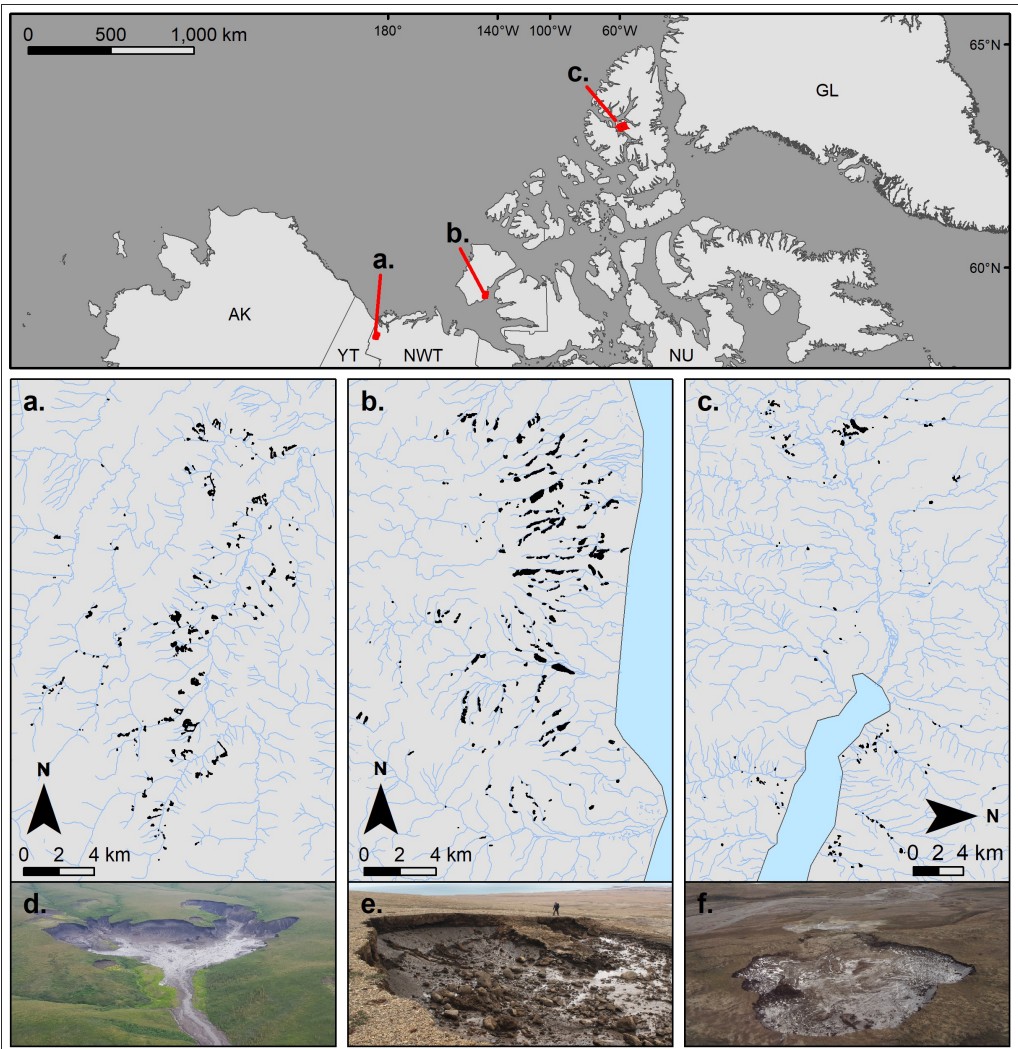

**Figure 1.** Map of the study area showing thaw slumps used as training data. (**a**) Willow River (WR) Area in the Foothills of the Richardson Mountains, (**b**) the Jesse Moraine (JM) on Eastern Banks Island, and (**c**) the Fosheim Peninsula (FP). The Oblique images in the bottom row show individual slumps in each area: (**d**) Willow River (Photo: Ashley Rudy), (**e**) Jesse Moraine (Photo: Ashley Rudy), and (**f**) the Fosheim Peninsula (Photo: Alison Cassidy).

In the Canadian Arctic, slump inventories compiled using manual digitization have guided research on heritage conservation [20], cumulative impacts [21], carbon and nitrogen cycling [22–26], and mercury contamination and mobility [27]. Unfortunately, slump impacted landscapes are evolving so rapidly that it has not been possible to maintain up-to-date inventories across the Arctic. Since these disturbances have the potential to impact transportation infrastructure [5,28] and human land use [29,30], there is pressing a need to improve systems for continuous detection and monitoring of these geohazards.

Automated techniques have been explored to meet the challenge of detection and monitoring of thaw slumps and other permafrost disturbances, but were limited to local sites or missed many small features. Several studies used temporally dense Landsat image stacks to identify thaw slumps [31–33] with machine learning algorithms such as random forest and support vector machine. Rudy et al. [34] used a semi-automated method to detect permafrost slope disturbances from multi-temporal high-resolution satellite (i.e., IKONOS) imagery. However, Landsat image stacks have a resolution of 30 m and miss many thaw slumps of small size (<100 m × 100 m) or in an early stage of development. The semi-automated method performed moderately well and detected 43% of permafrost slope disturbances in an area of ~20 km² but it is unlikely that it could be extended to a continental scale.

Advances in image processing and classification techniques based on deep learning are creating new opportunities to automate the detection and monitoring of thaw slumps across large areas using high-resolution remote sensing imagery. Deep learning approaches allow computational models to learn multi-level representations of data and have significantly improved the state-of-the-art in image processing [35]. Several recent studies have applied deep learning to permafrost areas to map thaw slumps [36,37] and ice-wedge polygons [38,39]. Huang et al. [36] used a convolutional neural network (CNN) approach to delineate thaw slumps in a homogenous study area on the Tibetan Plateau and achieved a high accuracy. Nitze et al. [37] compared the potential strengths and limitations of three deep learning algorithms used to map thaw slumps with PlanetScope CubeSat imagery across six Arctic regions. This study highlighted the importance of training data and limitations related to model transferability among regions. It is likely that similar methods can be used to inventory slumps across even larger geographic areas, but additional case studies are needed to test the generality of different approaches. Moreover, the balance between accuracy and efficiency has not been fully quantified, and potential solutions for improving the transferability need to be explored.

The objectives of this study are to assess the accuracy and efficiency of deep learning models for mapping thaw slumps and seek to improve their transferability across the Arctic. Specifically, we demonstrate the capability of the DeepLabv3+ deep learning model and compare its accuracy and efficiency with eight network architectures (i.e., backbones) using PlanetScope imagery across the Canadian Western Arctic. We evaluate the effectiveness of different data augmentation methods to improve model accuracies. We also evaluate the ability of a generative adversarial network (GAN) to create new training data and provide a means of domain adaptation to improving classifier transferability.

## 2. Study Areas

Three study areas in Canada's Arctic were selected for slump mapping tests: the Willow River (WR) catchment in the foothills of the Richardson Mountains, the Jesse Moraine (JM) on Eastern Banks Island, and the Fosheim Peninsula (FP) of Ellesmere Island (Figure 1). These areas span the Southern Arctic (WR) and Northern Arctic (JM and FP) terrestrial ecozones [40,41], traversing a distance of about 2000 km, and 12 degrees of latitude (68–80° N). The climate of all three study areas is characterized by long, cold winters and short summers, with mean annual temperatures ranging from −7 (WR) to −20 °C (FP). The vegetation of the Fosheim Peninsula and Jesse Moraine are strongly controlled by hydrology, with well-drained upland sites hosting partial cover (20–50 percent) of Salix-Dryas tundra. Poorly drained sites at the base of slopes or in valley bottoms are dominated by a nearly continuous cover of wet sedge meadow [41,42]. The vegetation of the Willow River is characterized by dwarf shrub and tussock tundra at high elevations, which transitions to upright shrub communities and open spruce woodlands at lower elevation [43]. WR and JM lie in fluvially incised, hummocky moraine, while FP contains fine-grained marine sediments that lie below the Holocene marine limit [43]. Sloping terrain and continuous permafrost with high ground ice content, including massive ice [19,44,45], provide optimal conditions for the development of retrogressive thaw slumps [46]. Increased thaw slump activity in the last decade at all three study areas has been attributed to a combination of recent warm summers and increasing rainfall [3,18,19,45]. Thaw slumps were mapped within areas covering 355 km$^2$ in WR, 350 km$^2$ in JM, and 1065 km$^2$ in FP.

## 3. Data and Methods

### 3.1. Satellite Images and Training Data

We downloaded PlanetScope images from Planet Labs (www.planet.com, accessed on 14 December 2020) via their Education and Research Program and chose the "Analytic_SR" product, which has been orthorectified and converted to surface reflectance. PlanetScope is a constellation of more than 100 CubeSats, which acquires daily images at a spatial resolution of approximately three meters. The 16-bit images contain four bands providing measurements in red, green, blue, and near-infrared wavelengths. All images for this

study were acquired in July and August of 2020 (Table 1). We obtained mosaic and daily PlanetScope images, and used them for different purposes. The mosaic images provided complete coverage for each study area and served as base maps for manually delineating thaw slump boundaries (i.e., ground truth). Daily PlanetScope images acquired on a specific date were used for testing the performance of the deep learning model under different image acquisition conditions (Figure 2). To utilize all four bands in DeepLabv3+, we converted each 4-band image to two 3-band images (RGB and nirGB) with a bit depth of 8 bits. RGB images combined red, green, and blue bands, while nirGB images included near-infrared, green, and blue bands. We also applied data augmentation to increase the size and diversity of training data. Specifically we used eight options: flip, blur, crop, scale, rotate, bright, contrast, and noise. We performed experiments to determine which combination of these options resulted in the highest mapping accuracy.

**Table 1.** PlanetScope images of the three study areas.

| Regions | Image Type | Acquisition Dates in 2020 |
|---|---|---|
| Willow River (WR) | Mosaic | 18 August |
| | Daily | 7, 8, 11, 13, 23, 70, 31 July; 18, 19, 24, 28, 29 August |
| Jesse Moraine (JM) | Mosaic | 20 August |
| | Daily | 5, 25 July; 11, 20 August |
| Fosheim Peninsula (FP) | Mosaic | 7 and 8 August |
| | Daily | 7, 17 July; 9 August |

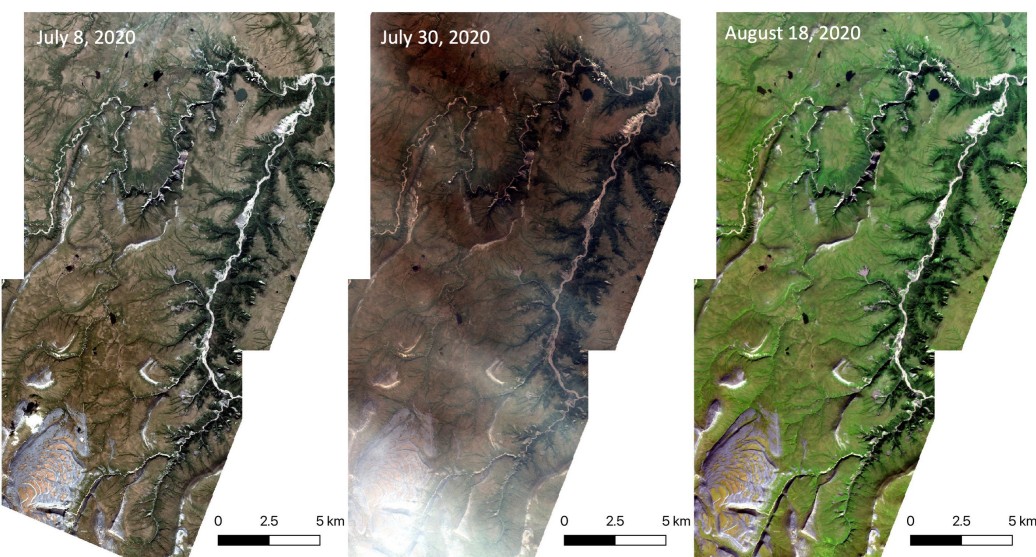

**Figure 2.** PlanetScope images (RGB) covering the Willow River (WR) demonstrating variable summer reflectance due to different illumination conditions, haze, and cloud shadow.

Training data were obtained by manually digitizing slumps using PlanetScope mosaic images. The final training polygons included thaw slumps (positive training data) and other features that appeared similar to slumps (negative training data) in each of the three study areas. Positive training data represents the boundaries of thaw slumps, and negative training data delineates areas of land cover similar to thaw slumps. A total of 197, 262, and 162 slumps were delineated within the WR, JM, and FP study regions, respectively. Two of our team members delineated and cross-checked the boundaries on mosaic images by inspecting each pixel in the three study areas and cross referencing with 50 cm resolution WorldView Imagery available through ESRI ArcGIS Online (www.arcgis.com, accessed on 16 December 2020). We initially ran a few classification iterations without negative training

polygons, then gradually added some negative polygons to address false positive results. We extracted sub-images from the PlanetScope images using a buffer size of 300 m and the corresponding raster label from the training polygons then tiled them with an overlap of 160 pixels to the size (<600 by 600 pixels) of the final training data [47].

### 3.2. Deep Learning Model

Automated slump delineation was completed using the DeepLabv3+ architecture [48]. This deep learning model with its state-of-the-art semantic segmentation algorithm has been shown to outperform many others in PASCAL VOC image segmentation tasks [49]. Semantic segmentation labels each pixel in an image, thereby mapping the location and extent of thaw slumps on satellite images. As shown in Figure 3, DeepLabv3+ consists of an encoder and decoder module. The encoder is built on popular CNN networks (termed as backbones) and Atrous Spatial Pyramid Pooling (ASPP). The CNN networks include Xception-41, 65, and 71 [50] , Mobilenetv 2 & 3 [51,52], and Resnet -50 and 101 [53] that were originally created for image classification (determine the image-level object shown in an image), but were repurposed for semantic segmentation by replacing the last few downsampling operators with ASPP. The numbers (e.g., 41) represent the depth of the corresponding network architectures. The decoder module recovers object segmentation details from low-level features and the encoder features. Eventually, for each input image, DeepLabv3+ outputs a labelled image (each pixel has a class id) of the same size.

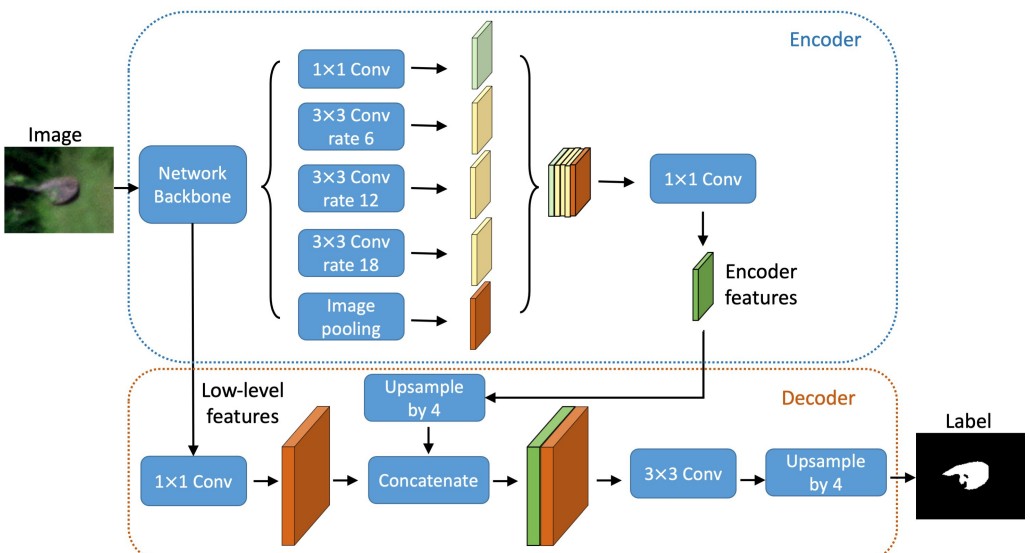

**Figure 3.** The architecture of Deeplabv3+ (modified from [48]).

### 3.3. Demonstrating the Capability of DeepLabv3+

To demonstrate the capability of DeepLabv3+ for mapping thaw slumps in several study areas, instead of only one homogeneous region [36] with available training data, we followed the procedure in [36] to delineate individual thaw slumps and calculate the precision, recall, and F1 score [36] for each region. The precision and recall were calculated after the post-processing of mapped polygons, and the threshold of polygon-based intersection over union for determining true positive was 0.5. We trained a DeepLabv3+ model using training data from the three areas with a learning rate of 0.014, a Xception65 backbone, and five data augmentation options (blur, crop, bright, contrast, and noise). These settings were derived from the experiments examining data augmentation, different backbones, and other hyper-parameters (see Section 3.4). We followed the steps detailed in [47] including tiling, predicting, and polygonizing, to generate polygons delineating the boundary of thaw slumps. In the post-processing step, we set a lower area threshold, taking into consideration the resolution of PlanetScope images (i.e., 3 m), to remove mapped polygons smaller than

900 m$^2$. We also utilized the topographic characteristics of thaw slumps to remove some false positives, as detailed in the next paragraph.

We removed mapped polygons with a slope smaller than one degree and those containing an area with decreasing elevation less than 16 m$^2$ by utilizing high-resolution (2 m) ArcticDEM elevation data [54]. We used the mosaic version of the ArcticDEM to calculate the mean slope within all slumps in the training data and used this to set a threshold for minimum slope (one degree) that effectively removed false positives. We did not set a threshold for maximum slope because terrain may have been disturbed before acquiring elevation data and shows a steep or even vertical headwall. False positives also were removed by calculating the changes in elevation over time using the strip version of ArcticDEM, which contains data from multiple time periods. We calculated the change in elevation using the oldest and most recent raster available for each pixel. Based on these calculations we generated binary rasters showing regions where elevation decreased (termed reduction zones) by setting a threshold of −0.5 m. For each mapped polygon, if there was a reduction zone greater than 16 m$^2$ within it or its 50-m buffer zone, we considered the polygon a thaw slump; otherwise, we deemed it to be a false positive.

### 3.4. Accessing the Accuracy and Efficiency of the Model

To assess the accuracy and efficiency of DeepLabv3+, we trained it with different backbones and other hyper-parameters by using PlanetScope images and training polygons derived from manual mapping. Instead of training the model from scratch, which would require significant computer resources and several weeks for each experiment, we fine-tuned the pre-trained models released by the DeepLab team (https://github.com/tensorflow/models/blob/master/research/deeplab/g3doc/model_zoo.md, accessed on 20 March 2020). We used the end-to-end training strategy to fine-tune the models, so all the layers would be fine-tuned. As reported by [48,55], model performance and efficiency vary significantly among different backbones. Therefore, we conducted experiments using different backbones and other hyper-parameters including learning rate and batch size for mapping thaw slumps. We used an open-source software package called Ray Tune [56] with a grid search strategy to run these experiments. For each experiment, we trained up to 30,000 iterations using 90% of the training data and retaining 10% for validation. During training, if model accuracy (measured using pIOU, as described in Equation (1)) did not improve after five evaluations, we stopped the training to prevent overfitting and save training time. After training, we used test data derived from images acquired on other dates or in other regions, but not included in the training data to evaluate model transferability. We used pixel-wise intersection over union (pIOU) to evaluate the accuracy of trained models:

$$pIOU = \frac{G \cap P}{G \cup P} \qquad (1)$$

where G is the ground truth label raster and P is the predicted raster. Since the model only includes two classes (thaw slump and background) in the semantic segmentation task, and their areas are highly imbalanced (slumps cover less than 10% of each study area), we only calculated the pIOU for thaw slump class as an indicator of model performance. Specifically, we used all pixels with the class of thaw slumps in G and P to calculate pIOU. This is different from many studies in which the mean value of pIOU is calculated across many classes (termed as mIOU). Unless stated otherwise, pIOU refers to the pixel-wise IOU of thaw slumps. pIOU assesses outputs from deep learning models and is not affected by any operations in the post-processing (Section 3.3) such as removing small mapped polygons. We ran two groups of experiments using different data augmentation options and hyper-parameters.

1. Data Augmentation Experiments. To save computing time, we restricted these experiments to the Willow River region and ran 255 (i.e., $\sum_{r=1}^{8} C(8, r)$) experiments using all possible combinations of the eight data augmentation options. Each option was used in 128 experiments.

2. Hyper-parameter Experiments. We also ran hyper-parameter experiments to explore trade-offs between model accuracy and efficiency. We merged the training data from the three study areas and applied data augmentation derived from group one (i.e., blur, crop, bright, contrast, and noise), then ran experiments of up to 30,000 iterations using different combinations of hyper-parameters including: (1) backbone (eight in total), (2) learning rates (0.007, 0.014, 0.021, 0.28), and (3) batch size (8, 16, 32, 48, 96), as these are three parameters that can significantly affect model accuracy and efficiency.

### 3.5. Using Domain Adaptation to Improving Transferability

To improve the transferability of the deep learning model for predicting thaw slumps using images acquired from different dates or regions, we adopted a generative adversarial network to generate new training data, as a method for domain adaptation. As creating training data for semantic segmentation is time consuming and requires expert mappers, domain adaptation has been used in many computer vision applications to tackle the problems of limited training data [57]. The goal of domain adaptation is to train algorithms on the data in the source domain and secure a good accuracy on the data in the target domain that is significantly different from the source [58].

Source and target domains differ among our study areas because: (1) different regions have different slump morphology, dominant vegetation, topography, and atmospheric conditions, (2) slump headwall expansion ranging from 1–100 m per year can significantly alter the extent of disturbances [28,59], and (3) PlanetScope images acquired on different dates within a single region are highly variable (Figure 2). We consider the mosaic images (Table 1) as the data in the source domains and daily images as the one the target domains. Usually, a GAN contains a generator and a discriminator that compete with each other during training [60]. The generator produces fake images showing scenes that do not exist, and the discriminator distinguishes fake between real images (i.e., training images). After adequate training, the discriminator should not be able to tell the difference between fake and real images. In this study we used a well-trained generator to produce images that are similar to the images in the target domains.

We chose images in the source and target domains and used CycleGAN to translate them from one domain to another, as shown in Figure 4. CycleGAN is a state-of-the-art GAN framework that can successfully translate images from one domain to another [61]. CycleGAN has two translating functions G: X->Y and F: Y->X (Figure 4) and uses two discriminators (not shown) to encourage G to produce G(X) images indistinguishable from domain Y and F(Y) images indistinguishable from domain X, respectively. To create source and target images, we extracted sub-images and the corresponding label raster, then used the sub-images and target images as training data for CycleGAN. We adopted the code published by the CycleGAN's authors and modified them for remote sensing data to emulate projection and tiling (github.com/yghlc/contrastive-unpaired-translation, accessed on 25 October 2021). We followed standard training protocol to train the CycleGAN up to 200 epochs as suggested by [61]. During training, we also monitored the output of CycleGAN and ensured that generated images appeared reasonable in the later stage of training. We used the well-trained translating function G to convert the sub-images into domain Y, then copied the corresponding label rasters, which created new training data that have the same distribution of domain Y and would be used to train DeepLabv3+.

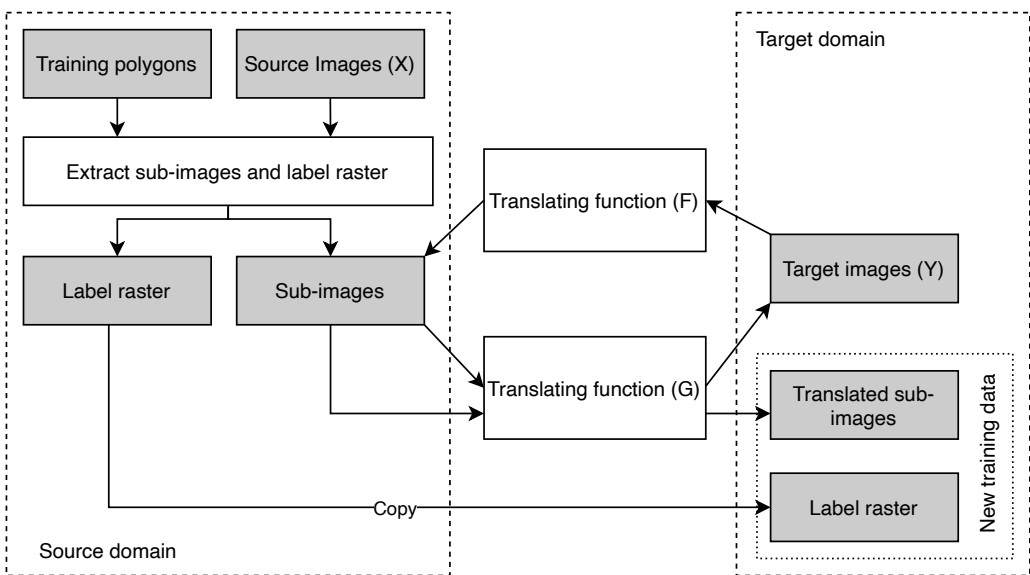

**Figure 4.** Flowchart showing the CycleGAN workflow used to generate new training data.

To evaluate the effectiveness of CycleGAN for domain adaptation and to understand its potential limitations, we conducted several experiments to test the temporal and spatial transferability of our deep learning model. For each study area, we conducted experiments to test the temporal transferability by using mosaic images as source images and other daily images as target images. To test the spatial transferability, we used the mosaic images of two study areas as source images and the remaining study area as target images. We also split each mosaic image into two roughly equal areas, then used half of the image as source images and the other half as target. In all experiments, we calculated the pIOU and compared the results with and without domain adaptation.

## 4. Results

### 4.1. Mapped Polygons of Thaw Slumps

The well-trained deep learning model (Section 3.3) accurately delineated the boundaries of thaw slumps and achieved a high F1 score. As shown in Figure 5, the deep-learning model delineated most thaw slump boundaries included in the ground truth data, indicating that the deep learning algorithm has the capability to represent thaw slump features in various regions. The F1 scores ranged from 0.676 to 0.849 for different images of the three study areas (Table 2). For each study area, the F1 score from nirGB images was higher than those from the RGB images, indicating that nirGB images are better for mapping thaw slumps in these regions.

**Table 2.** The accuracy of mapped polygons for three study areas (RGB and nirGB). The highest F1 score in each study region is shown in bold.

| Region (Image) | True Positive | False Positive | False Negative | Precision | Recall | F1-Score |
|---|---|---|---|---|---|---|
| Jesse Moraine (nirGB) | 219 | 35 | 43 | 0.862 | 0.836 | **0.849** |
| Jesse Moraine (RGB) | 206 | 46 | 56 | 0.817 | 0.786 | 0.802 |
| Fosheim Peninsula (nirGB) | 108 | 19 | 54 | 0.850 | 0.667 | **0.747** |
| Fosheim Peninsula (RGB) | 97 | 28 | 65 | 0.776 | 0.599 | 0.676 |
| Willow River (nirGB) | 154 | 29 | 43 | 0.842 | 0.782 | **0.811** |
| Willow River (RGB) | 148 | 28 | 49 | 0.841 | 0.751 | 0.794 |

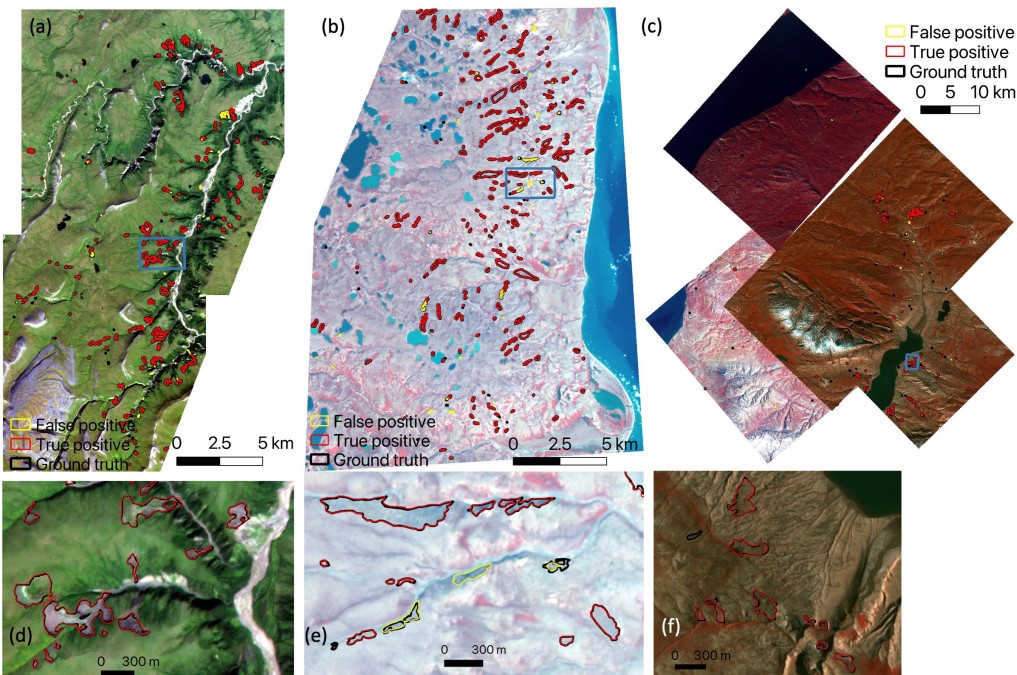

**Figure 5.** Mapped polygons in the three study areas. (**a**–**c**) are spatial distributions of mapped polygons for Willow River (WR), Jesse Moraine (JM), and Fosheim Peninsula (FP), respectively. (**d**–**f**) are enlargements of the regions in the blue rectangles in (**a**–**c**), respectively. The background of (**a**,**d**) are RGB images and others are nirGB images.

*4.2. Effectiveness of Different Data Augmentation Options*

The 255 trained models showed similar accuracy over the validation dataset, but had variable accuracy over the different test datasets. Each data augmentation option was used 128 times among the 255 experiments and achieved a similar mean (~0.83), maximum (~0.85), and minimum (~0.80) pIOU on the validation dataset derived from the WR region (Table 3). The maximum differences among the pIOUs of the eight different data augmentation options was within 0.001. This indicates that the trained model performs similarly on the validation dataset regardless of what combination of data augmentation options are used. Over the test datasets, the mean pIOU ranged from 0.06 to 0.78, while the max pIOU ranged from 0.20 to 0.79 (Table 4). The mean and maximum pIOU over the 0818 test dataset (#15 and #16 in Table 4) were close to the pIOU over validation dataset because the mosaic image in the WR region is also derived from images capture on 18 August. One interesting result is that the pIOU for models using nirGB images was consistently greater than those of the RGB ones, even though we trained the models after merging nirGB and RGB images. This suggests that the near-infrared band is more effective for identifying thaw slumps in the WR region where we completed this analysis (similar to Section 4.1).

**Table 3.** Accuracy of deep learning models using different data augmentation options. Table shows the mean, max, and min pIOU of validation dataset.

| Augmentation Option | Accuracy Statistics | | |
|---|---|---|---|
| | Mean pIOU | Max pIOU | Min pIOU |
| flip | 0.8263 | 0.845 | 0.7987 |
| blur | 0.8295 | 0.8499 | 0.8074 |
| crop | 0.8307 | 0.8499 | 0.8088 |
| scale | 0.8291 | 0.8486 | 0.7987 |
| rotate | 0.8222 | 0.8481 | 0.7984 |
| bright | 0.8282 | 0.8499 | 0.8074 |
| contrast | 0.8288 | 0.8499 | 0.7987 |
| noise | 0.8273 | 0.8499 | 0.7987 |

**Table 4.** Accuracy of deep learning models based on different test datasets. The table shows the mean and max pIOU of each test dataset when using the 255 models trained with different data augmentation options. The last column shows what data augmentation options were used in the model that achieved the maximum pIOU. Image acquisition dates are in the format of month day (e.g., 0708 means July 8).

| # | Image Dates | 3-Band | Mean pIOU | Max pIOU | Options Used (Max pIOU) |
|---|---|---|---|---|---|
| 1 | 0707 | nirGB | 0.4260 | 0.5018 | flip, blur, scale, bright, contrast, noise |
| 2 | | RGB | 0.1112 | 0.2685 | crop, contrast, noise |
| 3 | 0708 | nirGB | 0.5864 | 0.6741 | blur, crop, bright, contrast, noise |
| 4 | | RGB | 0.5746 | 0.6493 | blur, crop, scale, bright, noise |
| 5 | 0711 | nirGB | 0.5598 | 0.6100 | blur, crop, scale, bright, contrast, noise |
| 6 | | RGB | 0.5054 | 0.5677 | flip, blur, crop, bright |
| 7 | 0713 | nirGB | 0.4704 | 0.5382 | blur, scale, bright, contrast, noise |
| 8 | | RGB | 0.3214 | 0.4481 | crop, bright, noise |
| 9 | 0723 | nirGB | 0.6746 | 0.7232 | blur, crop, scale, contrast, noise |
| 10 | | RGB | 0.5630 | 0.6656 | crop, contrast, noise |
| 11 | 0730 | nirGB | 0.5669 | 0.6462 | blur, scale, bright, contrast, noise |
| 12 | | RGB | 0.1966 | 0.3542 | crop, contrast, noise |
| 13 | 0731 | nirGB | 0.5354 | 0.6010 | blur, rotate, bright, contrast, noise |
| 14 | | RGB | 0.0622 | 0.1980 | blur, crop, rotate, bright, noise |
| 15 | 0818 | nirGB | 0.7812 | 0.7927 | flip, blur, crop, scale, rotate, contrast |
| 16 | | RGB | 0.7759 | 0.7891 | flip, scale, contrast |
| 17 | 0819 | nirGB | 0.1740 | 0.3474 | flip, crop, rotate, bright, contrast, noise |
| 18 | 0824 | nirGB | 0.1169 | 0.2984 | crop |
| 19 | | RGB | 0.1301 | 0.2375 | flip, scale, bright, contrast, noise |
| 20 | 0828 | nirGB | 0.4578 | 0.5452 | crop, bright, contrast, noise |
| 21 | | RGB | 0.1773 | 0.2774 | crop, contrast, noise |
| 22 | 0829 | nirGB | 0.4875 | 0.5999 | crop, bright, contrast, noise |
| 23 | | RGB | 0.2636 | 0.3725 | crop, contrast, noise |

In data augmentation tests using imagery obtained on different days, the trained model that achieved the maximum pIOU used a different combination of data augmentation options, as shown in Table 4. Despite different data augmentation options among models, there were some that consistently contributed to the models that achieved maximum pIOU (Figure 6). For example, 'noise' was used 19 times while 'rotate' was only used 4 times, indicating that adding noise to training images is better than rotation for improving transferability. The top five options were 'noise' (19), 'contrast' (18), 'crop' (17), 'bright' (14), and 'blur' (11).

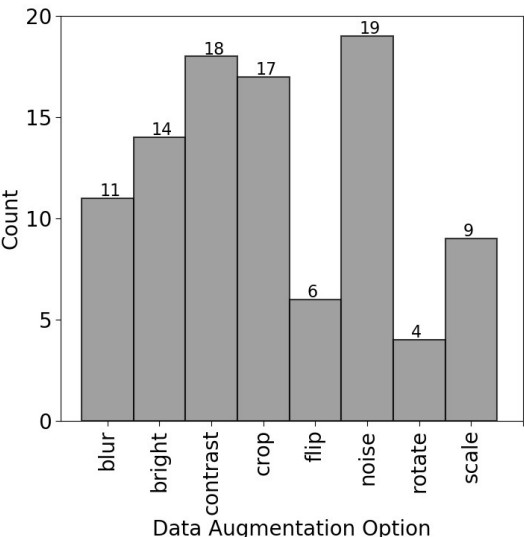

**Figure 6.** Frequency of each data augmentation option used in trained models reaching maximum pIOU in test datasets acquired on different dates (the last column in Table 4).

### 4.3. Accuracy and Efficiency When Using Different Hyper-Parameters

The accuracy, training time, and prediction time required when using different hyper-parameters were 0.675–0.844, 0.84–11.32 h, and 0.52–9.79 h, respectively (Figure 7 and Table 5). In total, 160 experiments using different combinations of backbones, learning rates, and batch size were performed but only 47 of these were successful because: (1) some experiments with inappropriate learning rates ended in divergent training loss and (2) some combinations were limited by GPU memory (32 GB, Tesla P100 × 2). The pIOU calculated using the validation dataset of these 47 experiments varied from 0.675 to 0.844, and training time required ranged from 0.84 and 11.32 h. The processing time required to make predictions across the three study areas ranged from 0.52 to 9.79 h (Table 5). Experiments with Xception achieved pIOU between 0.798 and 0.832, and but the larger depth of this network required more GPU memory (i.e., smaller batch size) and more training time (Figure 7a–c). A larger learning rate resulted in higher pIOU if the training was not divergent (Figure 7). A larger batch size generally required more training time but this did not guarantee improved pIOU (Figure 7b). Resnet also achieved a high pIOU, comparable to Xception, except for two experiments (Figure 7d,e). Compared with Resnet_v1_101, Resnet_v1_50 allowed larger batch size and achieved higher pIOU. Figure 7f,g show that MobileNet required less training time, but resulted in lower pIOU. MobileNetv3_large could also achieve pIOU that was similar to Xception and Resnet (Figure 7h).

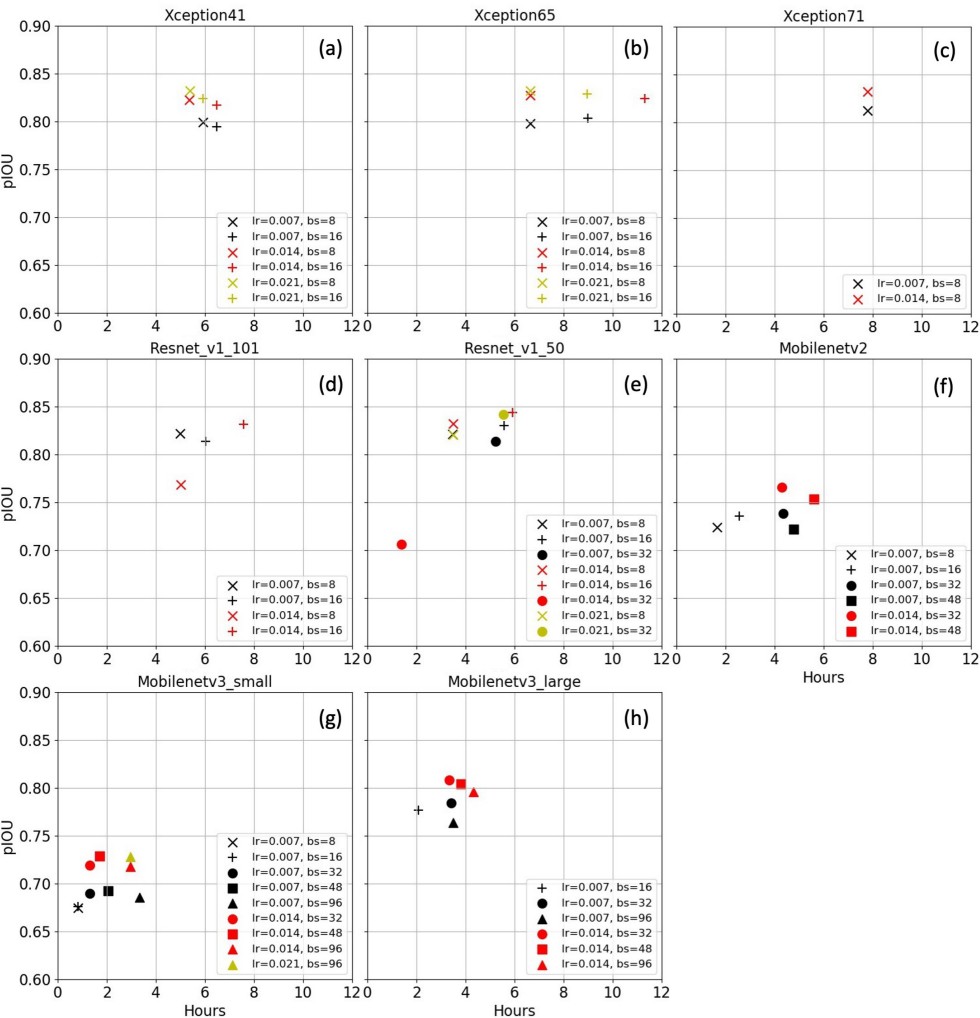

**Figure 7.** Accuracy of eight backbones and GPU time of different computational backbones used to run the DeepLab slump model. (**a**–**h**) show the accuracies of different backbones. Accuracy measured as pixel-wise intersection over union (pIOU) on the validation dataset and training time of eight backbones. "lr" is learning rate and "bs" is batch size. Y-axis is pIOU and X-axis hours.

**Table 5.** Computational time for prediction within three study areas using RGB and nirGB images.

| # | Backbone | Prediction Time (hours) |
|---|---|---|
| 1 | Xception41 | 6.78 |
| 2 | Xception65 | 9.29 |
| 3 | Xception71 | 9.79 |
| 4 | Resnet_v1_101 | 5.83 |
| 5 | Resnet_v1_50 | 3.82 |
| 6 | Mobilenetv2 | 1.79 |
| 7 | Mobilenetv3_large | 0.64 |
| 8 | Mobilenetv3_small | 0.52 |

### 4.4. Translated Images and Improvement Due to Domain Adaptation

Our experiments showed that CycleGAN can effectively translate images from one domain to another but may introduce some artifacts (Figure 8). Comparing the translated images in Figure 8b–d and PlanetScope images acquired on different dates in Figure 8e–g, showed that CycleGAN successfully translated the source image (Figure 8a) into the corresponding domains using different well-trained generators and indicates that CycleGAN has the capability for domain adaptation. However, it may also introduce artifacts as shown in Figure 8d whereby the strong texture within the extent of thaw slumps does not exist in the source or target images.

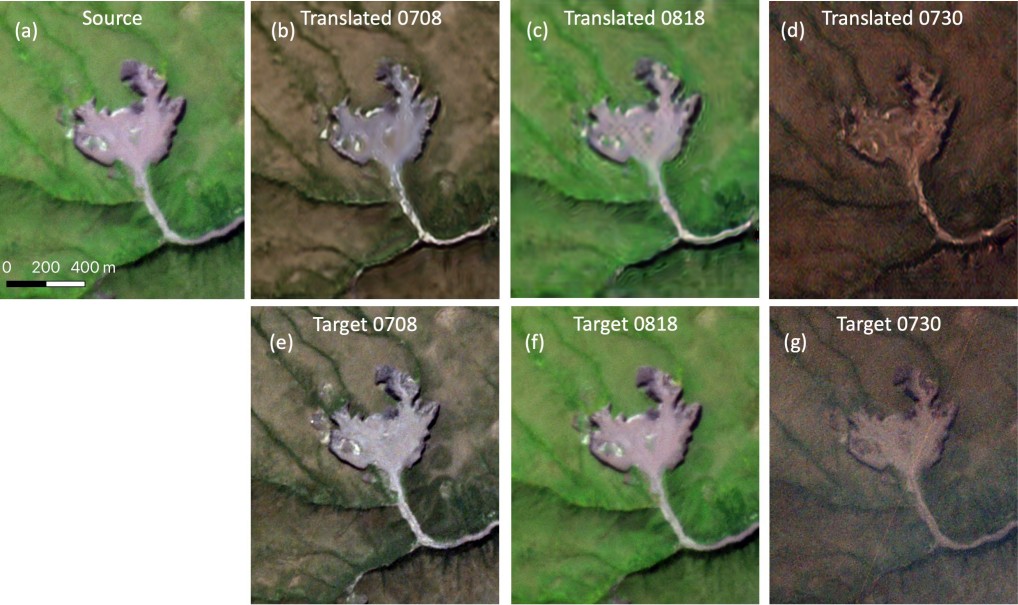

**Figure 8.** Source and translated images from the WR region: (**a**) is a sub-image from a source domain (the mosaic image), and (**b–d**) are translated images by different well-trained CycleGAN models. (**e–g**) are the corresponding target images. The numbers at the top or each image (e.g., 0708) show the acquisition dates of images in target domains.

When the difference between the source and target domain was too large, CycleGAN generated images that had similar color to target images but slump morphology was quite different (Figure 9). Comparisons of images from the source and target domains showed that the distance between these two domains was quite large, unlike the distance due to different acquisition dates in the same regions (Figure 8). Due to the large distance, CycleGAN produced translated images of thaw slumps (Figure 9c,d,g,h) that could be quite different from the features of thaw slumps in the target domain.

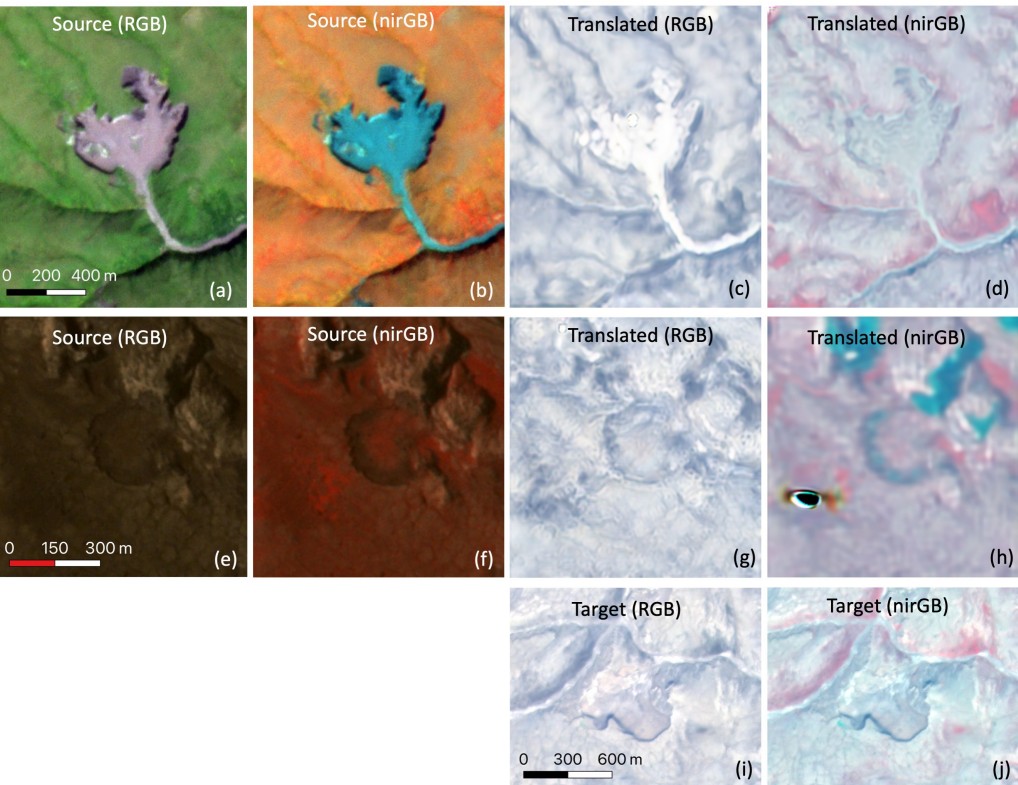

**Figure 9.** Sub-images from source (**a,b,e,f**) and target (**i,j**) domains, and translated sub-images (**c,d,g,h**). (**c,d**) were translated from (**a,b**), respectively, and (**g,h**) were translated from (**e,f**), respectively. The source domains are from two study areas (Willow River and Fosheim Peninsula) and the target domain is from the Jesse Morain.

Applying GAN for domain adaptation in experiments using images acquired on different dates resulted in some significant improvements in the pIOU. In the WR region, the improvement in pIOU with domain adaptation ranged from 0.01 to 0.55 (Figure 10a,d). The largest improvements in pIOU in these experiments involved images adapted from the target domain, while models using images from the source domain were largely unchanged. For example, the pIOU of mosaic images from August 18th remained the same or improved a little because these two are in the source domain, while other improvements were larger because they are in the target domains. In the JM and FP regions, domain adaptation resulted in similar improvements in pIOU (Figure 10b,c,e,f). A small improvement (<0.1) of pIOU for mosaic images in the three regions likely resulted from the larger training dataset when using CycleGAN to add new training data.

Experiments using two different study areas as the source domain and the remaining one as the target domain showed variable results, with both improvement and declines in accuracy. In the experiment of using the JM and FP as the source domain and the WR as target domain (Figure 11a,d), the pIOU for the RGB image declined by 0.09 (from 0.16 to 0.07), but the nirGB image showed an increase in accuracy of 0.26 (from 0.09 to 0.35). In the other two experiments (Figure 11b,c,e,f), all models showed negligible improvements in pIOU, except the JM region (Figure 11e). Experiments using different portions of images in one study area also showed that domain adaptation resulted in limited improvement (Figure 12). This is likely because the distance between two domains is quite large. Although CycleGAN can produce images that have similar color in the target domain, the thaw slumps in these images are morphologically or structurally distinct from the ones in the target domain, potentially confusing the deep learning model during training.

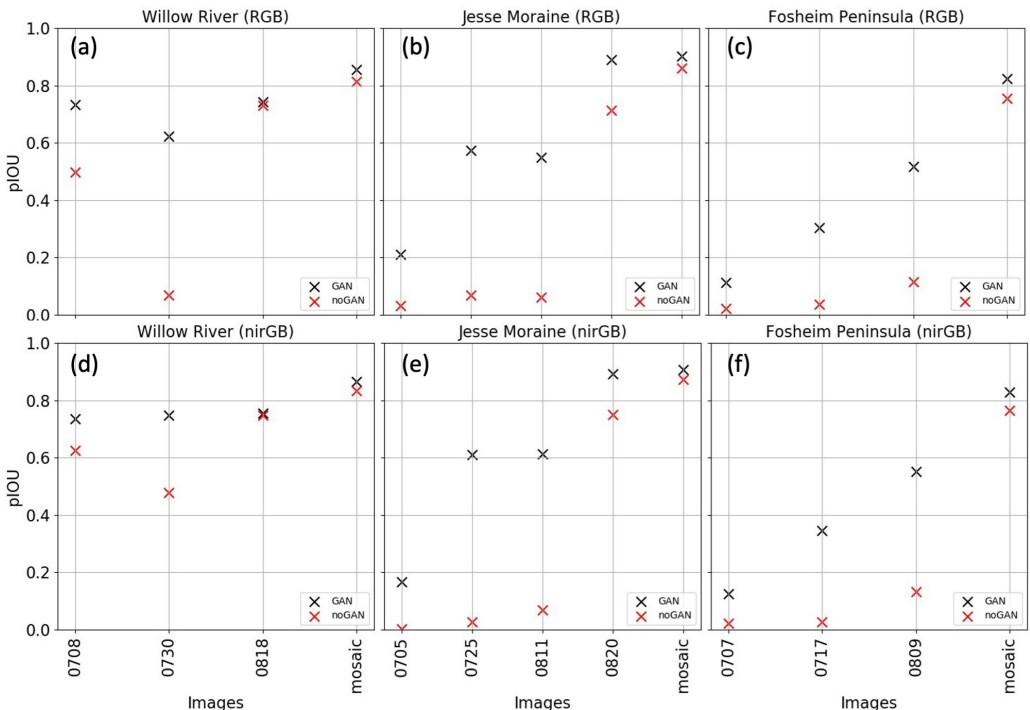

**Figure 10.** Comparison of the accuracy (pIOU) between experiments with and without GAN domain adaptation for images acquired on different dates. (**a**–**f**) show the accuracies of different regions using RGB and nirGB images. All images are from 2020 and the numbers on the x-axis (e.g., 0730) represent the acquisition month (e.g., 07) and day (e.g., 30).

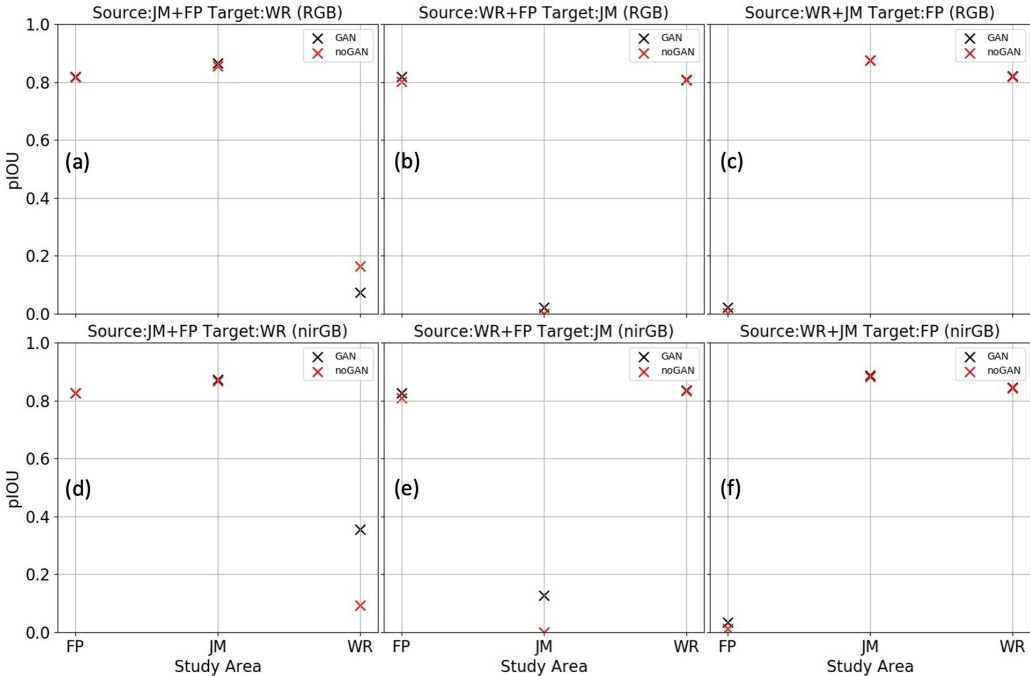

**Figure 11.** Comparison of the accuracy (pIOU) among experiments with and without GAN domain adaptation for images acquired in different study areas. (**a**–**f**) show the accuracies of different experiments. WR: Willow River, JM: Jesse Moraine, FP: Fosheim Peninsula.

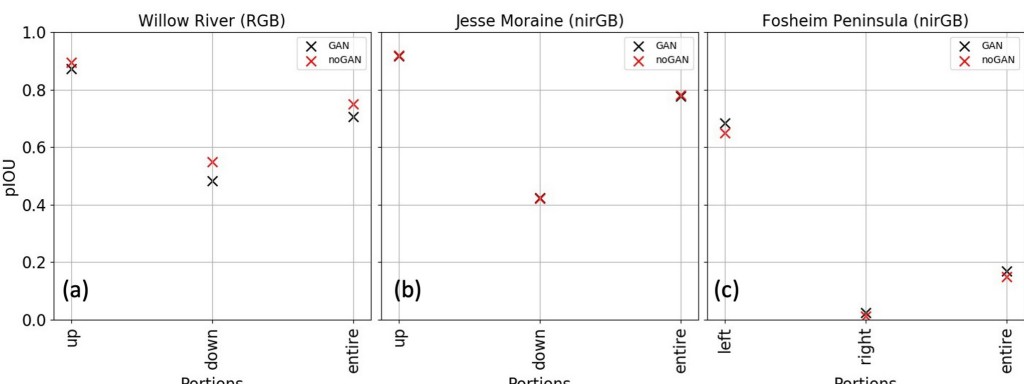

**Figure 12.** Comparison of the accuracy (pIOU) among experiments with and without GAN domain adaptation for different parts of the images in one region. (**a**–**c**) show the accuracies of different experiments.

## 5. Discussion

### 5.1. The Performance of the Deep Learning Model

DeepLabv3+ demonstrated strong capability to represent thaw slumps from diverse regions using images acquired under different conditions. We trained one model for three study areas and used it to delineate thaw slumps (Section 3.3). The results (Section 4.1) show that the model performs quite well, indicating that with sufficient training data deep learning can be used to map slumps at continental scales. As shown in [36], deep-learning-based boundaries closely matched manually mapped ones, suggesting that DeepLabv3+ can be used to delineate the boundaries of thaw slumps from multi-temporal images for change detection.

Our analysis shows that using different combinations of hyper-parameters (backbones, learning rates, and batch size, etc.) strongly affected both the model's accuracy and efficiency. High accuracy is typically a priority when mapping geohazards, but efficiency is also an important consideration when handling datasets covering large spatial extents. As presented in Figure 7, some of the backbones resulted in a smaller pIOU than others, but their efficiency was much higher (e.g., Mobilenetv3_large) owing to a simpler but less capable architecture. Therefore, the trade-off between accuracy and efficiency should be considered for any mapping application, especially when very large datasets must be processed.

Our analysis shows that transferability is a major challenge when applying deep learning models for slumps to new images acquired in different times or regions. For example, the trained model achieved pIOU of around 0.8 over the validation dataset (Table 3), but the pIOU ranged from around 0.1 to 0.8 (Table 4) when the model was applied to other images and other dates, despite the application of data augmentation to increase the volume and diversity of training data. In terms of spatial transferability, those models trained using data from two of our study areas to delineate slumps in the third all had pIOUs smaller than 0.16 or even zero, indicating that they could not be reliably applied to areas without local training data.

### 5.2. Strategies for Improving Transferability

A lack of training data is a common issue in deep learning applications, especially for mapping tasks in permafrost areas. Our analysis shows that data augmentation can improve the transferability of deep learning models in certain circumstances. Data augmentation can randomly increase the diversity of training data and balance examples among different classes, but does not guarantee improvements in transferability. As shown in Figure 6, some augmentation options such as noise and contrast that modify images by changing color or adding noise are more useful than others (e.g., rotate). A possible reason is that these options increased the diversity of image colors and allowed trained models to correctly classify thaw slumps in the test datasets that were acquired under different illumination and atmospheric conditions. This indicates that different domains require different data

augmentation options for improving the transferability or generalization on test data. It is also possible to automatically search for the best combination or strategy of data augmentation options within a designed space [62], but this requires expensive computing resources.

Another strategy for improving the transferability of deep learning models is to use GANs, but this may fail under certain circumstances. GANs can produce translated images that have similar features in target domains, such as thaw slumps in a new region or images acquired on a new date. During training, large differences in color or texture between source and target domains, including translated images, may improve the mapping accuracy. As shown in Figure 10, images for the same regions, but acquired on different dates, tend to fit this condition and are suitable for adopting a GAN for adding new training data. A GAN also may produce some unexpected, translated images when the distance between source and target domains is large in relation to slump color, texture, and morphology. For example, in Figure 9c, the translated color inside the slump is white although its surroundings are similar to what shown in Figure 9i. In Figure 9h, an anomalous black feature is produced by the GAN, and the slump also is modified significantly, especially the slump floor, making visual features of the slump unclear. These unnatural features or artifacts are not helpful for improving the transferability of deep learning models as shown in Figures 11 and 12. Instead, these unnatural features may confuse deep learning models and even lower their performance. Based on the results presented in Section 4.4, we conclude that GANs are a useful approach for improving temporal, but not spatial transferability.

*5.3. Mapping Practice with Deep Learning*

Our results suggest that deep learning models designed to map the boundaries of slumps face the fundamental challenge of spatial and temporal transferability. Given the impact that these issues have on model accuracy, we suggest that mapping initiatives covering multiple regions and time periods should focus on identification rather than delineation. Locating and delineating thaw slumps should be considered as two different objectives, although DeepLabv3+ can output both at the same time. When mapping across large spatial extents, the locations of unknown thaw slumps should be the priority of automated mapping. In this scenario, the main challenges include processing of large datasets and the transferability of the trained model due to variability in ecosystems and image properties. We recommend that analysts select automated mapping tools to fit the project objectives. For example, for continental-scale inventories, employing a backbone with high efficiency and good accuracy as shown in Figure 7 may be the best approach. In scenarios where the boundaries of thaw slumps are important, such as monitoring slump expansion, manual delineation may be a better approach if the number of thaw slumps is manageable.

A practical approach is necessary for handling false positives and false negatives from automated mapping algorithms. False positives and false negatives are always encountered. While the number of misclassifications can be relatively small, they can lead to uncertainties and concerns in any downstream analysis. Many factors can lead to false positives and negatives, including (1) uncertainties in the training data and ground truth, (2) imbalance between the classes among training data, and (3) the difference of data distribution between training data and images for prediction. From a technical perspective, reducing the number of false positives and negatives is the goal of many efforts in algorithm development. One practical approach that can reduce the number of false negatives is iterative mapping: performing automated mapping and manual validation in multiple iterations until no new thaw slumps are identified [63]. However, such efforts may not be necessary if the downstream analysis can tolerate some level of mapping uncertainty.

To apply deep learning models at a continental scale, data augmentation, GANs, and additional training data are likely to help overcome the challenge of transferability. Data augmentation should be the first approach to consider for improving transferability as it is the simplest and can be performed during training. However, it is likely that GANs will need to be combined with additional training data, as GANs can improve the transferability of deep learning models under certain conditions, while additional

training data are preferable if translating images covering different ecosystems is beyond the GAN's capabilities. For example, we suggest dividing the entire mapping region into many sub-regions based on their ecosystem types, collecting ground truth from some of the sub-regions, then conducting GAN experiments to test the transferability. Similar to the iterative mapping described in [63], this approach could be used to gradually collect additional training data from more sub-regions while running the GAN experiments. Using this strategy, we can build a deep learning model to map thaw slumps at the continental scale, and monitor changes in extent over time.

## 6. Conclusions

We conducted experiments to understand the accuracy, efficiency, and transferability of a deep learning model (i.e., DeepLabv3+) for mapping retrogressive thaw slumps in three, widely separated study regions in the Canadian Arctic. The results of our experiments show that: (1) different augmentation options can improve the size and diversity of training data, resulting in improved performance of the deep learning model, but may not be able to improve its transferability if the options do not narrow the distribution gap between training and test data; (2) different combinations of hyper-parameters (e.g., backbone, learning rate, and batch size) can affect both the accuracy and efficiency of the deep learning model; (3) some backbones (e.g., Mobilenetv3_large) have lower accuracy (i.e., pIOU) than others but much greater efficiency, suggesting that they would be suitable for large datasets; (4) using a GAN for domain adaptation can significantly improve the transferability of the deep learning model if the distance between source and target domain is within the color difference or changes of general texture. The deep learning model for mapping thaw slumps performs very well in regions with training data, but performance is variable when the model is applied to a new region. GANs provide an approach to overcome the issue of transferability, but may fail if the distance between source and target images is too large, suggesting that training data from the new region are necessary. Our findings provide guidance on how to collect training data for the purpose of global mapping. We suggest dividing large study regions into many sub-regions based on ecosystem type, and testing the transferability to determine which subregions require additional training data.

**Author Contributions:** Conceptualization, L.H., T.C.L., and R.H.F.; methodology, L.H.; software, L.H. and T.C.L.; formal analysis, L.H., T.C.L., and R.H.F.; resources, T.C.L., M.J.W., and K.F.T.; data curation, L.H. and T.C.L.; writing—original draft preparation, L.H., T.C.L., and R.H.F.; writing—review and editing, All; visualization, L.H. and T.C.L.; supervision, K.F.T., K.S., and M.J.W. All authors have read and agreed to the published version of the manuscript.

**Funding:** Lingcao Huang was supported by the CIRES Visiting Fellows Program and the NOAA Cooperative Agreement with CIRES, NA17OAR4320101. Financial support was also provided by the NWT Cumulative Impact Monitoring Program, the Natural Sciences and Engineering Research Council of Canada (PermafrostNet and RGPIN 06210-2018: T.C.L.), and NASA Grant (NNX17AC59A: K.S.).

**Data Availability Statement:** PlanetScope images can be acquired through www.planet.com. Manually-delineated polygons of retrogressive thaw slumps may be provided by Trevor C. Lantz upon request. The setting files and codes are available at https://github.com/yghlc/Landuse_DL (all accessed on 27 April 2022).

**Acknowledgments:** The authors would like to thank Mike Newton for compiling the training data for this study and assistance in preparing Figure 1. We would like to thank Planet's Education and Research Program, through which we obtained PlanetScope Images for this study.

**Conflicts of Interest:** The authors declare no conflict of interest. The funders had no role in the design of the study; in the collection, analyses, or interpretation of data; in the writing of the manuscript, or in the decision to publish the results.

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
