# Peer review of "Accuracy, Efficiency, and Transferability of a Deep Learning Model for Mapping Retrogressive Thaw Slumps across the Canadian Arctic"

_remotesensing, doi:10.3390/rs14122747_

Round 1

Reviewer 1 Report

Thanks for the submission.

Ethical Concern: Author 2 has intentionally self-cited 7-8 papers which is a very poor practice in Academia.  

Comments:

In this study, Deep learning has been used for mapping retrogressive thaw slumps and other periglacial landforms but its application is still limited to local study areas

Supplementary Material's MUST be provided to verify the claims author have made.

I can't see explanation of GAN architecture and its limitations with reference to current problem in particular. Just journal claims are being made. 

Please add in the 2nd last paragraph of your introduction – regarding aim of this study and objectives.

For a study like this, it is important to separately add some recent literature Section, Also focusing on how did you conducted keywords search and mention relevant sources (Google Scholars, WoS, Scopus) - so that readers can get better idea of relevant literature in this domain.

Conclusion section can be improved by highlighting key findings, limitation of the research and recommendation for future studies.

Methods used in the study needs to be compared well with the efficiency of one method over the other being compared statistically, I can see that missing clearly.

Explain why the current method was selected for the study, its importance and compare with traditional methods.

Also add risk factors in your study – risk associated with this domain – risk matrix. / False positive and True negative should be added. 

Please pay special attention to how the methodology and the methods used are presented. As it is currently presented, the reader does not get a clear picture of what analysis was done in the research.

Another issue is the lack of a clear message about a novelty in the study. There is a need for a clear message of what the authors have elaborated.

Reviewer 2 Report

The article is well written and is of interest to the journal. It may be published in this form.

Author Response

Thank you!

Reviewer 3 Report

This paper evaluates the performance of the deep learning model in the mapping of retrogressive thaw slumps in polar regions, and gives the way to improve the transferability of the deep learning model. The research of this paper is of positive significance to the large-scale mapping of Cryosphere surface evolution. It is suggested that the authors make a more detailed introduction to the methods and the way to get  data and images, so as to be used for reference by readers.

Reviewer 4 Report

This manuscript assessed the accuracy, efficiency, and transferability of deep learning models for mapping retrogressive thaw slumps, an important periglacial landforms in cold region. The manuscript is well structured. The data and methods are also clearly described, including information regarding satellite images and training data, deep learning model, experiments design and accuracy assessment. The results of the study is also informative for mapping thaw slumps over large spatial extents using deep learning. Because of this I believe that this paper deserves to be published in Remote sensing. 
I just have three mini suggestions. 
1. Introduction. What is the situation of other non-deep learning model for identifying the thaw slumps? You origin focus on deep learning model can be enhanced in Introduction section.
2. That is the more information are needed for the metrics such as recall and F1-score in section 3.3 (Line 132). 
3. Line 371-373. This sentence is not clear for me. 

Round 2

Reviewer 1 Report

Thank you for providing with the feedback. Just a gentle suggestion that be polite while responding to the comments made or concerns raised.

Acceptance and rejection is very common in Academia but it doesn't mean you offensive or rude with reviewers who are just trying to help you improve the quality of your work. 

I can see the claims made by authors are valid but authors didn't address all the concerns raised. 

Good Luck!